# A High Resolution Mass Spectrometry Study Reveals the Potential of Disulfide Formation in Human Mitochondrial Voltage-Dependent Anion Selective Channel Isoforms (hVDACs)

**DOI:** 10.3390/ijms21041468

**Published:** 2020-02-21

**Authors:** Maria G. G. Pittalà, Rosaria Saletti, Simona Reina, Vincenzo Cunsolo, Vito De Pinto, Salvatore Foti

**Affiliations:** 1Department of Biomedical and Biotechnological Sciences, University of Catania, Via S. Sofia 64, 95123 Catania, Italy; marinella.pitt@virgilio.it; 2Department of Chemical Sciences, Organic Mass Spectrometry Laboratory, University of Catania, Viale A. Doria 6, 95125 Catania, Italy; vcunsolo@unict.it (V.C.); sfoti@unict.it (S.F.); 3Department of Biological, Geological and Environmental Sciences, Section of Molecular Biology, University of Catania, Viale A. Doria 6, 95125 Catania, Italy; simonareina@yahoo.it

**Keywords:** Cysteine over-oxidation, mitochondria, Orbitrap Fusion Tribrid, hydroxyapatite, mitochondrial intermembrane space, outer mitochondrial membrane, post-translational modification

## Abstract

The voltage-dependent anion-selective channels (VDACs), which are also known as eukaryotic porins, are pore-forming proteins, which allow for the passage of ions and small molecules across the outer mitochondrial membrane (OMM). They are involved in complex interactions that regulate organelle and cellular metabolism. We have recently reported the post-translational modifications (PTMs) of the three VDAC isoforms purified from rat liver mitochondria (rVDACs), showing, for the first time, the over-oxidation of the cysteine residues as an exclusive feature of VDACs. Noteworthy, this peculiar PTM is not detectable in other integral membrane mitochondrial proteins, as defined by their elution at low salt concentration by a hydroxyapatite column. In this study, the association of tryptic and chymotryptic proteolysis with UHPLC/High Resolution nESI-MS/MS, allowed for us to extend the investigation to the human VDACs. The over-oxidation of the cysteine residues, essentially irreversible in cell conditions, was as also certained in VDAC isoforms from human cells. In human VDAC2 and 3 isoforms the permanently reduced state of a cluster of close cysteines indicates the possibility that disulfide bridges are formed in the proteins. Importantly, the detailed oxidative PTMs that are found in human VDACs confirm and sustain our previous findings in rat tissues, claiming for a predictable characterization that has to be conveyed in the functional role of VDAC proteins within the cell. Data are available via ProteomeXchange with identifier PXD017482.

## 1. Introduction

The voltage-dependent anion selective channels (VDACs) are quantitatively relevant components of the mitochondrial proteome. Located in the outer mitochondrial membrane of all eukaryotes, VDACs form aqueous pores that govern ions and metabolites exchange across the organelle, thus contributing to cellular homeostasis maintenance. VDAC has emerged as a key player in cancer [1] and in neurodegenerative diseases, such as Alzheimer’s disease (AD), Parkinson’s disease (PD), and Amyotrophic Lateral Sclerosis (ALS), because of its central role in metabolism and its interaction with several cytosolic enzymes and apoptotic factors [2].
In higher eukaryotes three VDAC isoforms, named VDAC1, VDAC2, and VDAC3, which are encoded by three separate genes located on different chromosomes, have been characterized [3]. VDAC1 is the most abundant and ubiquitously expressed of the three isoforms, while VDAC2 and VDAC3 have been estimated to be 10 and 100 times less expressed, respectively. The evolutionary history of VDACs suggests VDAC3 as the oldest isoform, while VDAC1/2 should derive from its duplication around 350 million years ago; VDAC1 is considered to be the youngest isoform since it diverged from VDAC2 300 million years ago. In mammals, VDAC1 and VDAC3 genes contain nine exons, while VDAC2 10: the additional exon encodes a 11 N-terminal amino acids extension with undetermined functional significance. VDAC isoforms display roughly 75% nucleotide sequence identity. Many mitochondrial porin sequences have been obtained by recombinant DNA technology [4,5,6,7]. VDAC2 from bovine spermatozoa and VDACs from rat liver mitochondria (rVDAC) have been partially characterized [8]. Interestingly, 90% sequence identity between the human and rat isoforms can be appreciated: in particular, the number and position of cysteine residues among VDACs of these mammal species are highly conserved. Rat and human VDAC2 and VDAC3, for example, both contain cysteines in the N-terminal α-helix, whereas VDAC1 has none. The evolutionary preservation of these residues could suggest a role of cysteine in the protein functionality.
We were able to fully cover the rVDAC1 sequence, and extended the coverage of rVDAC2 and 3, by means of a newly established “in-solution” enzymatic proteolysis and UHPLC/High Resolution ESI MS/MS analysis in a previous work, which was performed on rat-derived mitochondria. This procedure enabled to avoid most of the possible technical biases in the protocol that could have modified the naturally occurring oxidation state of the oxidizable residues. We could then propose a detailed profile of the oxidation state of methionine and cysteine residues and the identification of other post-translational modifications for all three rat VDAC isoforms [9,10,11]. We also demonstrated that cysteine over-oxidation is peculiar to VDACs, since it was not observed in other highly hydrophobic transmembrane mitochondrial proteins analyzed. Here, the same experimental approach was applied to human VDACs that were purified from HAP1 cells mitochondria, producing the respective other post-translational modifications pattern. The results indicate that 1) the oxidative state of cysteine residues is conserved between rat and human; 2) there is a specific utilization of each specific cysteine, since some equivalent residues are always found in the reduced form (i.e., free thiol groups that are available to the oxidation to disulfide bridges), while others are constantly irreversibly over-oxidized as sulfonic acid. The consequences of these results are discussed.

## 2. Results

### 2.1. Mass Spectrometry Analysis of Human VDAC1

Reduction/alkylation of proteins was carried out before VDACs purification from the mitochondria that were obtained from HAP1 cells to exclude the possibility that any unspecific and/or undesired oxidation could happen during the purification protocol. Hydroxyapatite (HTP) eluates of Triton X-100 extract were in-solution digested by trypsin and hereinafter analyzed by mass spectrometry. In this experiment, every protein in the HTP eluate was digested, thus a very complex peptide mixture was produced. Appendix A reports hVDAC1 tryptic peptides that were identified in the analysis of mitochondria lysate incubated with dithiothreitol (DTT). UHPLC/High Resolution nESI-MS/MS of the tryptic digest allowed to almost completely cover the reported protein sequence (279 of 282 amino acid residues), with the exception of the tripeptide N^111^AK^113^ (Figure 1 and Appendix A). Although some predicted tryptic peptides of the hVDAC1 sequence are shared by two or three isoforms, unequivocal sequence coverage was obtained by the detection of unique peptides originated by missed cleavages whose sequences are boxed in Figure 1.

The results also confirmed that the N-terminal Met, as reported in the SwissProt database sequence (Acc. N. P21796), is missing in the mature protein and that the N-terminal Ala is acetylated (Figure 1, Appendix A, molecular fragments 1, 2, and 3).

We particularly focused our investigations on the oxidation state of Met and Cys residues. The sequence of hVDAC1 contains two methionines in positions 129 and 155, and two cysteines in positions 127 and 232 (following the sequence numbering, including the starting Met^1^, which is actually absent in the mature protein). Met^129^ and Met^155^ were detected in peptides in the normal form (Appendix A, fragment 1, and Appendix A, fragments 22 and 23), and also in the form of Met sulfoxide (Appendix A, fragments 2 and 3). Figure 2 and Figure 3, respectively, report the full scan and fragment ion mass spectra of the doubly charged molecular ion of the peptide E^121^HINLGCDMDFDIAGPSIR^139^ with Met^129^ modified as methionine sulfoxide and Cys^127^ in the form of sulfonic acid, and of the peptide G^140^ALVLGYEGWLAGYQMNFETAK^161^ with Met^155^ as methionine sulfoxide. The MS/MS spectra both show the typical neutral loss of 64 Da corresponding to the ejection of methanesulfenic acid from the side chain of MetO [12].

Although from these data a precise determination of the ratio between Met and Met sulfoxide cannot be obtained, a rough estimation of their relative abundance can be derived from the comparison of the absolute intensity of the multiply charged molecular ions of the respective peptides (Table 1). These calculations (Table 1) indicate approximately the same ratio of MetO/Met for the two methionines (1.5:1 for Met^129^, and 0.8:1 MetO/Met for Met^155^, respectively).

Analysis of the mass spectral data that were oriented to the determination of the oxidation state of cysteines showed that Cys^232^ was totally found in the carboxyamidomethylated form (Appendix A, fragment 32), while Cys^127^ was exclusively detected in the oxidized form of sulfonic acid (Figure 2, Appendix A, fragments 1 and 2).

Chymotrypsin also digested hVDAC1. Complementary data obtained from the chymotryptic fragments resulted in the complete coverage of the sequence, including the stretch N^111^AK^113^, (Appendix A, fragment 19). This analysis also confirmed the lack of Met^1^, the acetylation of Ala^2^ (Appendix A, fragments 1 and 2), and the partial oxidation of Met^129^ to methionine sulfoxide (Appendix A, fragments 1-3). In addition, Cys^127^ was only observed in trioxidized form (Appendix A, fragments 1–3) and Cys^232^ exclusively in the carboxyamidomethylated form (Appendix A, fragments 30 and 31), as found in the tryptic digest.

### 2.2. Mass Spectrometry Analysis of Human VDAC2

The peptides that were identified by tryptic digestion and UHPLC/High Resolution nESI-MS/MS of hVDAC2 covered 97.3% of the hVDAC2 sequence (285 of 293 amino acid residues) (Figure 4, Appendix A). Analogously to the hVDAC1 isoform, the N-terminal methionine, as reported in the SwissProt database sequence (Acc. N. P45880), is absent and the Ala^2^ is in the acetylated form (Appendix A, fragment 1).

Additionally, hVDAC2 shows tryptic peptides sequences that are shared by one or two other isoforms: unequivocal sequence coverage was obtained by detecting unique peptides originated by missed cleavages. Figure 4 boxes the sequences of these unique peptides.

In the hVDAC2 there are two internal methionines in position 12 and 166, and nine cysteines (sequence positions: 8, 13, 47, 76, 103, 133, 138, 210, and 227). Met^12^ and Met^166^ were identified both as normal Met (Appendix A, fragment 2 and Appendix A, fragment 6) and as Met sulfoxide (Appendix A, fragments 1, 2, and 7). The comparison of the absolute intensity of the doubly or triply charged molecular ions of the corresponding fragments indicates a ratio of approximately 10:1 and 1.5:1 MetO/Met for Met^12^ and Met^166^, respectively (Table 2). Instead, the triply charged molecular ion of the tryptic peptide E^132^-K^172^, including two cysteines and Met^166^ was detected, but its corresponding MS/MS spectrum was not obtained: thus, Met^166^ was found either in the normal (Appendix A, fragment 6) and oxidized form (Appendix A, fragment 7), while it was not possible to determine which one of the two cysteines (Cys^133^ or Cys^138^) was oxidized or reduced.

Moreover, the partial oxidation of Cys^47^, Cys^76^, Cys^103^, and Cys^210^ to sulfonic acid (Appendix A, fragments 3, 4, 5, and 8, respectively, and Appendix A), and the presence of the cysteines 8, 13, and 227 exclusively in the carboxyamidomethylated form were revealed (Appendix A, fragments 1, 2, and 21). We could roughly estimate the relative abundance of the cysteines oxidized to sulfonic acid with respect to those in the carboxyamidomethylated form: this estimation is related with the ratio of the absolute intensity of the multiply charged molecular ions of the corresponding peptides, as reported in Table 2. In the evaluation of these results, it should be considered that the oxidation of cysteine to cysteine sulfonic acid introduces a strong negative charge in the peptide, thus hampering the formation of positive ions. However, even while taking these considerations into account, the data in Table 2 suggest that measurable amounts of Cys^76^ and Cys^210^ are oxidized to sulfonic acid (ratio ox/red 0.1), whereas Cys^47^ and Cys^103^ oxidized to sulfonic acid are only present in trace amounts (ratio ox/red 0.01 and 0.04, respectively).

Chymotryptic digestion of hVDAC2 was also performed to increase the sequence coverage. Combining the results that wereobtained in the tryptic and chymotryptic digestions, the coverage of the sequence of hVDAC2 was extended to 98,3% (288 out of 293 amino acid residues) (Figure 3 and Appendix A). The region that is still not covered corresponds to the sequence Ser^173^-Arg^177^.

Analysis of the chymotryptic digest confirmed the presence of cysteines 13, and 227 exclusively in the carboxyamidomethylated form (Appendix A, fragments 1, 2, and 29), and of Cys^210^ in the normal and trioxidized form (Appendix A, fragment 27, and Appendix A, fragment 3).

Two fragments containing Cys^133^ and Cys^138^ were present in the chymotryptic mixture (Appendix A, fragments 1 and 2). The MS/MS spectrum of the doubly charged molecular ion of fragment 1 (Appendix A) demonstrated that Cys^133^ was totally carboxyamidomethylated whereas Cys^138^ was completely in the form of sulfonic acid, thus allowing to characterize the oxidation state of these two cysteines that was not possible to determine in the analysis of the tryptic digest (see above). At the end, chymotryptic peptides, including methionines 12 and 166, were undetectable.

### 2.3. Mass Spectrometry Analysis of Human VDAC3

UHPLC/nESI-MS/MS of the in-solution tryptic digest of hVDAC3 resulted in a coverage of about 91% of the sequence (256 of 282 amino acid residues, Appendix A and Figure 5), thus extending the coverage that was obtained by other Authors from in-gel digestion and MALDI-ToF/ToF mass spectrometric analysis [13]. Although some predicted tryptic peptides of the hVDAC3 sequence are shared by other isoforms, unequivocal attribution for two of them was obtained by the detection of unique peptides originated by missed cleavages (Appendix A, fragments 6 and 19, and Figure 5).

Similar to the other two isoforms, it was found that the N-terminal Met, as reported in the SwissProt database sequence (Acc. N. Q9Y277), is removed in the mature protein, and that the N-terminal Cys is present in the acetylated form (Appendix A, fragment 1).

hVDAC3 sequence includes three methionines in position 26, 155, and 226. For all of these three methionines, peptides containing residues in the normal state (Appendix A, fragments 2, 4, 5, 12, and 17, respectively) and also oxidized to methionine sulfoxide were detected (Appendix A, fragments 1, 4, and 5), with a ratio of about 0.1:1, 1:1, and 3:1 MetO/Met, respectively (Appendix A and Table 3).

In hVDAC3, there are six cysteines in position 2, 8, 36, 65, 122, and 229. Analysis of the mass spectral data showed that Cys^2^ and Cys^8^, of which the N-terminal Cys^2^ was acetylated, were exclusively in the reduced and carboxyamidomethylated form (Appendix A, fragment 1). Also cysteines 122 and 229 (Appendix A, fragments 12 and 17) were only identified in the reduced form. Cysteines 36 and 65 were found both in peptides containing these residues carboxyamidomethylated (Appendix A, fragments 6, 7, and 9), and in peptides containing these residues in the form of sulfonic acid (Appendix A, fragments 2 and 3, respectively). From the comparison of the peak areas of the double charged molecular ions of the corresponding peptides, it was estimated that the ratio Ox/Red for these two cysteines is about 0.1–0.2 (Table 3).

The chymotryptic digestion of hVDAC3 was also performed. Although this analysis resulted in a low sequence coverage (63.5%, Appendix A), the complementary data obtained allowed to extend the coverage to 94.3% (266 out of 282 amino acid residues). It should be noted that the peptide G^220^IAAKY^225^ is common to all of the isoforms and, therefore, its attribution to the hVDAC3 sequence it is not univocal (Appendix A, fragment 22). The regions not verified yet correspond to the sequences Ile^29^-Leu^31^, Ser^111^-Arg^120^, and Thr^231^-Leu^233^ (Figure 5).

In addition, the data that were obtained confirmed the absence of Met^1^, the acetylation of Cys^2^, and the presence of Cys^2^ and Cys^8^ exclusively in the carboxyamidomethylated form (Appendix A, fragments 1 and 2).

## 3. Discussion

The characterization of VDACs presents challenging issues due to extremely high hydrophobicity, the difficulty of isolating each single isoform, and of separating them from other mitochondrial proteins of similar hydrophobicity. Consequently, it is necessary to analyze them as components of a relatively complex mixture. Furthermore, SDS-electrophoretic separation, which could be used for a preliminary isolation, should desirably be avoided due to the potential danger of unwanted oxidation reactions, which could introduce artifacts in the proteins. For these reasons, we have developed an “in-solution” digestion procedure that can be directly applied to the enriched VDAC isoforms fraction obtained as eluate from hydroxyapatite chromatography. The resulting proteolytic mixture can be subsequently analyzed by UHLC/High-Resolution mass spectrometry. We applied this procedure to enriched protein extracts of rat liver mitochondria, with the aim of characterizing the VDAC isoforms PTMs of sulfur amino acids [10,11]. We obtained wide information about them (Table 4) with the surprising result of specific oxidations to sulfonic acid. In this work, we extended this analysis to human VDAC isoforms that were obtained by enriched mitochondrial protein extracts from cultured cells. The aim was to extend our observation to the human proteins, to validate them, and verify their specificity. Using our "in-solution" protocol, in the present work we have been able to completely cover the sequence of hVDAC1, and almost completely the sequences of hVDAC2 and hVDAC3 (98.3% and 94.3%, respectively) from HAP1 cell mitochondria. It should be noted that the short regions not identified in hVDAC2 and hVDAC3 do not contain cysteines and methionines and they correspond to small tryptic or chymotryptic peptides or even to single amino acids, which cannot be detected in LC/MS analysis. In conclusion, the data obtained allowed for the determination of the oxidative states of all the cysteine and methionine residues present in the human isoforms.

### 3.1. Oxidation States of Methionines in Human and Rat VDAC Isoforms

As previously found for the three rVDACs [10,11] also in the hVDAC isoforms methionines oxidized to sulfoxide and cysteines over-oxidized up to sulfonic acid were observed. Most internal methionines are conserved between rat and human notwithstanding the N-terminal methionines, usually cleaved up in every VDAC isoforms maturation process. In the sequence of rVDAC1 one methionine residue (Met^155^) is conserved in hVDAC1. On the other hand, the rVDAC1 conserved Met^155^ is oxidized to MetO in a remarkable higher amount (Ox/Red ratio 65:1) than in the hVDAC1 (Ox/Red ratio 1:1). rVDAC2 shows only one internal Met (Met^167^). The rat Met^167^ and the human conserved homologous Met^166^ are both oxidized to MetO in approximately equal amount (Ox/Red ratio 2:1 in hVDAC2 and 3:1 in rVDAC2). Met^12^, exclusively found in hVDAC2, is mainly in the oxidized form (Ox/Red ratio 10:1). hVDAC3 sequence includes Met^26^, Met^155^, and Met^226^. For all these three methionines, the ratio between oxidized (methionine sulfoxide)/normal was found to be approximately 0.1:1, 1:1, and 3:1. Met^26^ in hVDAC3 showed an oxidation state that was comparable to the analogous methionine in rVDAC3. In rVDAC3, the oxidation rate of Met^155^, the latter conserved methionine, was not determined.

Methionine residues are highly susceptible to oxidation, even in mild conditions: reactive oxygen (ROS) and nitrogen species (NOS) generate in vivo methionine sulfoxide (MetO). MetO can be reduced back to methionine *in vivo* by ubiquitous sulfoxide reductases [14]. Cyclic Met oxidation/MetO reduction leads to the consumption of ROS and it is considered as a scavenging system from oxidative damage [15,16]. On the other hand, MetO formation has little or no effect on protein susceptibility to proteolytic degradation [17,18].

### 3.2. Oxidation States of Cysteines in Human and Rat VDAC Isoforms

Cysteine oxidation has structural relevance for proteins, since it can support the disulfide bridge formation, a physiological protein cross-linking. An excess of oxidation, in contrast, can be irreversible and even harmful in the cell, producing a permanent modification to sulfone or sulfine with the fixation of permanent negative charges in the protein [19]. Thus, we examined with the highest attention the fate of cysteine residues in the human VDAC isoforms and compared their state with those of rat VDAC isoforms [10,11].

VDAC1 shows two cysteine residues that were conserved between rat and human (Table 4). In rat VDAC1 both cysteines are moderately oxidized, with a ratio Ox/Red in the range of 0.1–0.2 [11]. Instead in human VDAC1, Cys^127^, which is embedded in the hydrophobic moiety of the outer mitochondrial membrane (OMM) (Figure 6/Appendix A), was found always and completely trioxidized, while Cys^232^, facing the aqueous inside of the pore, was exclusively observed in the carboxyamidomethylated form, an indication that, in the native protein, it is available to reversible oxidation to form disulfide bridges.

VDAC2 is the longer isoform, with an N-terminal extension of 11 amino acids in mammals. It is also the richest in cysteines. In rat, VDAC2 shows 11 cysteines, while human VDAC2, only nine (Table 4). This difference is due to the presence in the rat N-terminal of two additional cysteines. Additionally, in hVDAC2, some Cys were found always completely reduced, as in rVDAC2, with no trace of over-oxidation. Other Cys were instead detected in both chemical states (Table 4). In particular, Cys^8^, Cys^13^, and Cys^133^ were exclusively found in the reduced form: these residues are exposed to the IMS and correspond to Cys^9^, Cys^14^, and Cys^134^ in rVDAC2 that have also been found to be reduced or not determined (Cys^134^) [11]. Cys^227^, which is also exposed to the cytosol, was found exclusively in the reduced form analogously to the Cys^228^ of rVDAC2, whose trioxidized form was detected in very low amount [11]. In rVDAC2, also the additional Cys^4^ and Cys^5^, in the N-terminal moiety, are reduced; thus, VDAC2 has a cluster of cysteines exposed to the IMS that are available to reversible oxidation.

The oxidation state of the other positions is essentially conserved between rat and human. For hVDAC2, Cys^138^, located in a position similar to Cys^127^ in hVDAC1, and therefore embedded in the hydrophobic phase of OMM, was totally found in the form of sulfonic acid (Figure 6/Appendix A). The sequence containing the corresponding residue in rVDAC2 (Cys^139^) was not covered in our previous work and, consequently, its oxidation state could not be determined [11]. Assuming that, also in rat, this residue is over-oxidized, it is striking to notice that both VDAC1 and VDAC2 have an over-oxidized (and negatively charged) cysteine residue that is embedded in the hydrophobic milieu of the outer membrane. Such a peculiar, potentially destabilizing situation, lacks in the VDAC3, where this residue is absent, and could have a biological meaning that is possibly related to the different functions of the isoforms.

Cysteines in position 47, 76, 103, and 210 are in the ring region that is exposed to inter-membrane space (IMS). Among them, Cys^76^ was partially detected in the form of sulfonic acid with an Ox/Red ratio of about 0.1:1, similarly to the homologue Cys^77^ in rVDAC2 [11]. Cys^47^, Cys^103^ and Cys^210^ are also found partially trioxidized but with a lower Ox/Red ratio (about 0.1:1 – 0.01:1), reproducing the trend observed for the homologous Cys^48^, Cys^104^, and Cys^211^ of rVDAC2 [11].

Among the six cysteines contained in the hVDAC3 sequence, Cys^36^ and Cys^65^ were detected trioxidized with an Ox/Red ratio of 0.2:1 and 0.1:1, respectively. Homologous Cys^36^ and Cys^65^ in rVDAC3 were found to be oxidized to a similar extend (Ox/Red ratio of 1.4 and 0.2, respectively). Cys^2^ and Cys^122^, which are also facing the IMS (Figure 6/Appendix A), were found only in the reduced form, as Cys^8^, which is inside the channel. The same result was obtained for the corresponding Cys^2^ and Cys^8^ of rVDAC3. The oxidation state of Cys^122^ of the rVDAC3 remained undetermined, because peptides containing this residue were not detectable, but, in analogy with what found for Cys^122^ in hVDAC3 and Cys^133^ in hVDAC2, it is presumably in the reduced form. By contrast, Cys^229^ in hVDAC3 was completely in the reduced form, whereas the same residue was totally trioxidized in rVDAC3.

### 3.3. Sulfur Oxidation State of VDAC Isoforms is Conserved Between Rat and Human, and Between Tissues and Cultured Cells

The comparison of the oxidation rate of methionines and cysteines among the human VDAC isoforms with the corresponding rat VDAC isoforms provides some interesting information. As a preliminary observation, it is important to underline that most of the data obtained for the oxidation states of cysteines in human VDACs are the same as those reported for the rat VDACs, thus demonstrating that they do not depend on the type of organism, at least in mammals, but they reflect a physiological condition. The peculiarity of the over-oxidation of these mitochondrial proteins has been reported in our previous works, where we analyzed several integral mitochondrial membrane proteins, but we did not discover a relevant amount of over-oxidized cysteines, if any, in other proteins [11]. Another validation of these data is that the application of the same methodologies to different kind of sources, living tissue from organism (rat liver), or cultivated cells (human HAP cell) resulted in remarkably similar results.

The pattern of cysteine oxidation of VDAC isoforms might foresee structural and functional consequences. We noticed that there are different classes of cysteines with respect to their location in the protein and to their oxidation state (Figure 6).

1. A first class of cysteines is those whose lateral residue protrudes and that are embedded in the hydrophobic moiety of the outer membrane. These are Cys^127^ in VDAC1 and Cys^138^ in VDAC2. Both were only found in the sulfonic, tri-oxidized state: the most hydrophilic, the most sterically bulk, the most improbable for a presence in a hydrophobic milieu. We have no idea about this particular position, but we notice that VDAC1 and VDAC2 also possess the well-known Glu^73^ residue (Figure 1 and Figure 4), another hydrophilic residue that protrudes toward the hydrophobic part of the membrane, being involved in the binding of hexokinase to VDAC [20,21,22] and possibly in the oligomerization of the protein [23]. Interestingly, VDAC3 does not show any residue homologous to the Cys^127/138^ or the Glu^73^. This could mean that the isoform 1 and 2 have a different role in the stabilization/destabilization of the membrane or have relationships with different phospholipids.

2. A second class includes the abundant number of cysteines in the N-terminal sequence. There are two cysteines in the short sequences of the first amino acids in both VDAC2 and VDAC3 and they are even four in the homologous sequence of rat VDAC2. They were found always in the carboxymethylated form, suggesting that they are reduced in the cell or in a reversibly oxidized fashion. The most obvious of these states are disulfide bridges. We can also add Cys^122^ in VDAC3 and Cys^133/134^ in VDAC2 as members of this class, since they have also been found as stably carboxyamidomethylated, and they are in close proximity to the N-terminal cysteines, as shown in the structural model of VDAC2 and VDAC3 built upon the hVDAC1 available three-dimensional (3D) structure [22,24,25].

3. The third group of cysteines comprises a ring of residues predicted to be exposed to the IMS. These residues (Cys^47^, Cys^76^, Cys^103^, and Cys^210^ in hVDAC2 and Cys^36^ and Cys^65^ in hVDAC3) were found either in the oxidized or in the reduced (carboxyamidomethylated) forms, in variable ratios: they thus look to be residues subject to variable and possible reversible oxidations. These last residues would form the reservoir of the VDACs as a buffer against the oxidative action of ROS and other similar agents.

A less defined role has to be assigned to Cys^227/228^ in VDAC2 and Cys^229^ in VDAC3. The latter is present in a predicted turn that was exposed to the IMS. It was found always reduced in human, but always oxidized in rat: it is perhaps another member of the ring of cysteines proposed to buffer the oxidative potential of the IMS. Cys^227/228^ of VDAC2 is, instead, the only cysteine predicted to be in a hydrophilic turn exposed to the cytosol. Its position is intriguing, also because it has been found always in reduced form in human, and almost completely reduced in rat. It is tempting to speculate that it can be involved in some kind of docking function of this protein.

## 4. Materials and Methods

### 4.1. Chemicals

All of the chemicals were of the highest purity commercially available and used without further purification. Ammonium bicarbonate, calcium chloride, phosphate buffered saline (PBS), Tris-HCl, Triton X-100, sucrose, mannitol, ethylene glycol tetraacetic acid (EGTA), ethylenediaminetetraacetic acid (EDTA), 4-(2-hydroxyethyl)-1-piperazineethanesulfonic acid (HEPES), formic acid (FA), dithiothreitol (DTT), iodoacetamide (IAA), and fetal bovine serum (FBS) were obtained from Sigma–Aldrich (Milan, Italy). IMDM (Iscove’s Modified Dulbecco’s Medium containing L-glutamine and 25 mM HEPES) was obtained from Gibco-Thermo Fisher Scientific. Human HAP1 cells were from Horizon Discovery (Cambridge, United Kingdom UK). Modified porcine trypsin and chymotrypsin were purchased from Promega (Milan, Italy). Water and acetonitrile (OPTIMA® LC/MS grade) for LC/MS analyses were provided from Fisher Scientific (Milan, Italy).

### 4.2. Extraction of Proteins from Human HAP1 Cell Culture Mitochondria Under Reducing Condition

Human HAP1 cells were cultured in monolayer (75 cm^2^ tissue culture flask) until 75% confluence while using IMDM that was supplemented with 10% FBS and 1% penicillin/streptomycin (Invitrogen, Carlsbad, CA, USA). The cells were maintained in a humidified incubator at 37 °C with 5% CO_2_. For mitochondria preparation, HAP1 cells were harvested by EDTA and washed with PBS three times before disruption. Every PBS wash was eliminated by centrifugation at 1500× *g* for five minutes at 4 °C [26].

The total cell pellet that was obtained was resuspended in 1 ml of hypotonic buffer (200 mM mannitol, 70 mM sucrose, 10 mM HEPES pH 7.5, 1 mM EGTA pH 8.0). The cells were incubated in ice for 10 min and then lysed. The lysate obtained was centrifuged (700× *g* for 25 min at 4 °C) to eliminate the non-lysed cells and the nuclei. To have a better yield, after recovering the supernatant, the pellet containing the mitochondria, was again suspended in hypotonic buffer. Again, the suspension was first lysed and then centrifuged (700× *g* for 25 min at 4 °C).

The resulting supernatants from the two centrifugations were combined and then centrifuged at higher speed (7000× *g* for 15 min at 4 °C). The supernatant, containing the cytoplasmic fraction, was removed, while the pellet was washed with hypotonic buffer. The suspension was then centrifuged at high speed (7000× *g* for 15 min at 4 °C) and at the end, after removing the supernatant, the pellet containing the mitochondria was resuspended in 500 μL of hypotonic buffer and stored at 4 °C.

The total yield was determined by micro-Lowry method, resulting 3.65 g/L. The hypotonic buffer was then removed by centrifugation (10,000× *g* for 20 min at 4 °C). Reduction/alkylation was carried on before VDACs purification to avoid any possible artefact due to air exposure. 1.83 mg protein of intact mitochondria were incubated for 3 h at 4 °C in 1 ml of Tris-HCl 10 mM (pH 8.3) containing 0.037 mmol of DTT, which corresponds to an excess of 50:1 (mol/mol) over the estimated protein thiol groups. The temperature was kept at 4 °C to avoid possible reduction of methionine sulfoxide to methionine by methionine sulfoxide reductase. After centrifugation for 30 min at 10,000× *g* at 4 °C, alkylation was performed by addition of IAA at the 2:1M ratio over DTT for 1 h in the dark at 25 °C. The mixture was centrifuged for 30 min at 10,000× *g* at 4 °C and pellet was stored at −80 °C until further use.

The reduced and alkylated intact mitochondria were lysed in buffer A (10 mM Tris-HCl, 1 mM EDTA, 3% Triton X-100, pH 7.0) at a ratio 5:1 (mitochondria mg/buffer volume mL) [27] for 30 min. on ice and centrifuged at 17,400× *g* for 30 min. at 4 °C. The supernatant containing mitochondrial proteins was subsequently loaded onto a homemade glass column 5 × 80 mm, packed with 0.6 g of dry hydroxyapatite (Bio-Gel HTP, Biorad, Milan, Italy) [27]. The column was eluted with buffer A at 4 °C and fractions of 500 μL were collected and tested for protein content by a fluorometer assay (Invitrogen Qubit™ Protein Assay kit, ThermoFisher Scientific, Milan, Italy). Fractions containing proteins were pooled and dried under vacuum. The hydroxyapatite eluate was divided in two aliquots, which were reduced to less than 100 µL and then purified from non-protein contaminating molecules with the PlusOne 2-D Clean-Up kit (GE Healthcare Life Sciences, Milan, Italy), according to the manufacturer’s instructions. The desalted protein pellet was suspended in 100 μL of 50 mM ammonium bicarbonate (pH 8.3) and incubated at 4 °C for 15 min. Next, 100 μL of 0.2% RapiGest SF (Waters, Milan, Italy) in 50 mM ammonium bicarbonate were added and the samples were kept at 4 °C for 30 min. Another aliquot of desalted protein pellet was suspended in 100 μL of 100 mM Tris-HCl, 10 mM calcium chloride (pH 8.0) and then incubated at 4 °C for 15 min. Next, 100 μL of 0.2% RapiGest SF in 100 mM Tris-HCl, 10 mM calcium chloride, were added and the samples were kept at 4 °C for 30 min. For each fraction, the recovered protein amount was determined in 40 μg by using a fluorometer assay. Reduced and alkylated proteins were then separately subjected to digestion while using modified porcine trypsin and chymotrypsin, respectively. Tryptic digestion was carried out at an enzyme-substrate ratio of 1:50 at 37 °C for 4 h; chymotryptic digestion was performed in Tris-HCl 100 mM, 10mM calcium chloride (pH 8.0) at an enzyme-substrate ratio of 1:20 overnight at 25 °C.

### 4.3. Liquid Chromatography and Tandem Mass Spectrometry (LC–MS/MS) Analysis

Mass spectrometry data were acquired on an Orbitrap Fusion Tribrid (Q-OT-qIT) mass spectrometer (ThermoFisher Scientific, Bremen, Germany) that was equipped with a ThermoFisher Scientific Dionex UltiMate 3000 RSLCnano system (Sunnyvale, CA, USA). Samples that were obtained by in-solution tryptic and chymotryptic digestion were reconstituted in 100 μL of 5% FA aqueous solution and 1 μL was loaded onto an Acclaim®Nano Trap C18 column (100 μm i.d.×2 cm, 5 μm particle size, 100 Å). After washing the trapping column with solvent A (H_2_O + 0.1% FA) for 3 min. at a flow rate of 7 μL/min., the peptides were eluted from the trapping column onto a PepMap® RSLC C18 EASYSpray, 75 μm × 50 cm, 2 μm, 100 Å column and were separated by elution at a flow rate of 0.250 μL/min., at 40 °C, with a linear gradient of solvent B (CH_3_CN + 0.1% FA) in A from 5% to 20% in 32 min., followed by 20% to 40% in 30 min., and 40% to 60% in 20 min. The eluted peptides were ionized by a nanospray (Easy-spray ion source, Thermo Scientific) while using a spray voltage of 1.7 kV and introduced into the mass spectrometer through a heated ion transfer tube (275 °C). The survey scans of peptide precursors in the m/z range 400–1600 were performed at resolution of 120,000 (@ 200 m/z) with a AGC target for Orbitrap survey of 4.0 × 10^5^ and a maximum injection time of 50 ms. Tandem MS was performed by isolation at 1.6 Th with the quadrupole, and high energy collisional dissociation (HCD) was performed in the Ion Routing Multipole (IRM), while using a normalized collision energy of 35 and rapid scan MS analysis in the ion trap. Only those precursors with charge state 1–3 and intensity above the threshold of 5·10^3^ were sampled for MS^2^. The dynamic exclusion duration was set to 60 s with a 10 ppm tolerance around the selected precursor and its isotopes. Monoisotopic precursor selection was turned on. The AGC target and maximum injection time (ms) for MS/MS spectra were 1.0 × 10^4^ and 100, respectively. The instrument was run in top speed mode with 3 s cycles, meaning that the instrument would continuously perform MS^2^ events until the list of non-excluded precursors diminishes to zero or 3 s, whichever is shorter. MS/MS spectral quality was enhanced, enabling the parallelizable time option (i.e., by using all parallelizable time during full scan detection for MS/MS precursor injection and detection). Mass spectrometer calibration was performed while using the Pierce® LTQ Velos ESI Positive Ion Calibration Solution (Thermo Fisher Scientific). MS data acquisition was performed using the Xcalibur v. 3.0.63 software (Thermo Fisher Scientific).

### 4.4. Database Search

The LC–MS/MS data were processed by PEAKS software v. 8.5 (Bioinformatics Solutions Inc., Waterloo, ON, Canada). The data were searched against the 49,544 entries “Human” SwissProt database (release June 2019) Full tryptic or chymotrypsin peptides with a maximum of three missed cleavage sites were subjected to bioinformatic search. Cysteine carboxyamidomethylation was set as fixed modification, whereas the acetylation of protein N-terminal, trioxidation of cysteine, oxidation of methionine, and transformation of N-terminal glutamine and N-terminal glutamic acid residue in the pyroglutamic acid form were included as variable modifications. The precursor mass tolerance threshold was 10 ppm and the max fragment mass error was set to 0.6 Da. Finally, all of the protein hits obtained were processed by using the *inChorus* function of PEAKS. This tool combines the database search results of PEAKS software with those that were obtained by the Mascot search engine with the aim not only to increase the coverage, but also the confidence, since the engines using independent algorithms and, therefore, confirm each other’s’ results. Peptide spectral matches (PSM) were validated while using Target Decoy PSM Validator node based on q-values at a 0.1% False Discovery Rate (FDR).

The mass spectrometry proteomics data have been deposited to the ProteomeXchange Consortium (http://proteomecentral.proteomexchange.org) via the PRIDE partner repository [28] with the dataset identifier PXD017482.

## 5. Conclusions

The panoramic view of sulfur amino acids modification state of VDAC in mammals provides new, exciting perspectives for future researches aiming to unravel the role of each VDAC isoform in the control of OMM permeability, in the response to the ROS oxidative pressure on IMS, and, in general, on the mitochondrial quality control. From a technical point of view, the identification of disulfide bridge(s) is the next, highly challenging goal of our research.

## Figures and Tables

**Figure 1 ijms-21-01468-f001:**
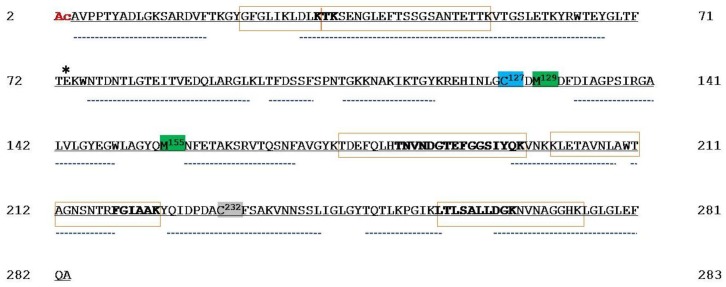
Sequence coverage map of hVDAC1 obtained by tryptic and chymotryptic digestion. The gray solid line indicates the sequence that was obtained with tryptic peptides; dotted lines the sequence obtained with chymotryptic peptides. The carboxyamidomethylated cysteine residue is highlighted in grey, the totally oxidized cysteine residue is highlighted in light blue and the partially oxidized methionine residues are highlighted in green. The acetyl group linked to the N-terminal alanine is shown in red. Unique tryptic peptides originated by missed-cleavages were used to distinguish and cover sequences shared by isoforms: they are indicated in the boxes. Sequences shared by more isoforms are reported in bold. Numbering of the sequence considered the starting methionine residue, actually deleted during protein maturation. Aspartic acid residue 73 shared by hVDAC1 and 2 is marked by an asterisk.

**Figure 2 ijms-21-01468-f002:**
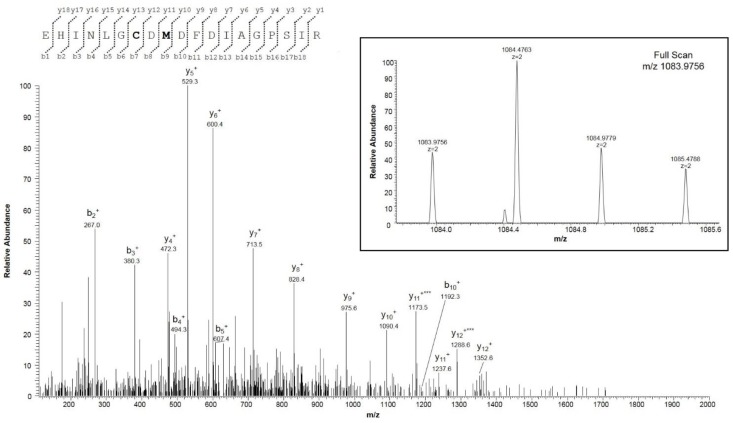
MS/MS spectrum of the doubly charged molecular ion at m/z 1083.9756 (calculated 1083.9754) of the VDAC1 tryptic peptide from HAP1 cells containing cysteine residue 127 in the form of sulfonic acid and methionine residue 129 in the oxidized form of methionine sulfoxide. The inset shows the full scan mass spectrum of molecular ion. Fragment ions originated from the neutral loss of methanesulfenic acid (CH_2_SOH, 64 Da) are indicated by three asterisks.

**Figure 3 ijms-21-01468-f003:**
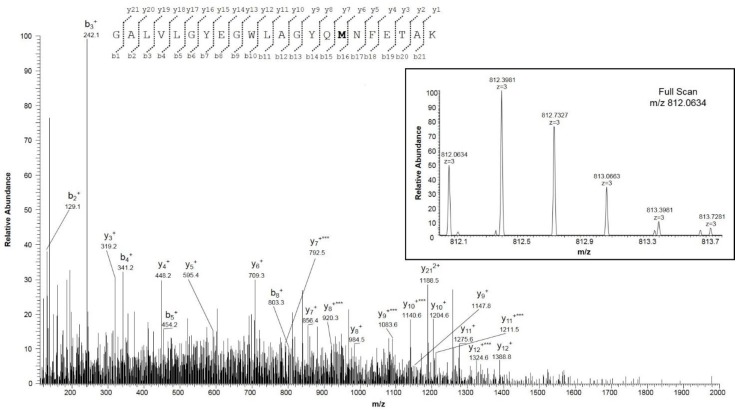
MS/MS spectrum of the triply charged molecular ion at m/z 812.0634 (calculated 812.0635) of the VDAC1 tryptic peptide from HAP1 cells containing methionine residue 155 in the oxidized form of methionine sulfoxide. The inset shows the full scan mass spectrum of molecular ion. Fragment ions originated from the neutral loss of methanesulfenic acid (CH_2_SOH, 64 Da) are indicated by three asterisks.

**Figure 4 ijms-21-01468-f004:**
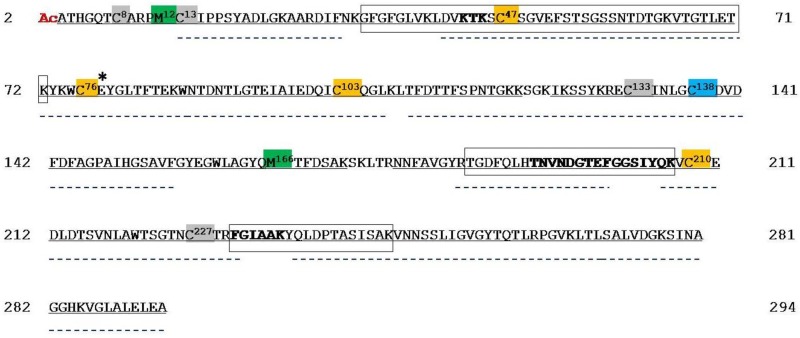
Sequence coverage map of hVDAC2 obtained by tryptic and chymotryptic digestion. Gray solid lines indicate the sequence obtained with tryptic peptides; dotted lines the sequence obtained with chymotryptic peptides. The carboxyamidomethylated cysteine residues are highlighted in grey, the partially oxidized cysteine residues are highlighted in orange, and the totally oxidized alanine residue is highlighted in light blue. The partially oxidized methionine residues are highlighted in green. The acetyl group linked to the N-terminal cysteine is shown in red. Unique tryptic peptides originated by missed-cleavages were used to distinguish and cover sequences shared by isoforms: they are indicated in the boxes. Sequences shared by isoforms are reported in bold. Numbering of the sequence considered the starting methionine residue, actually deleted during protein maturation. Aspartic acid residue 77 shared by hVDAC1 and 2 is marked by an asterisk.

**Figure 5 ijms-21-01468-f005:**
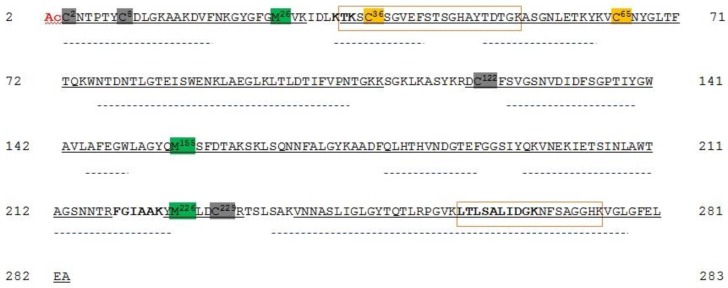
Sequence coverage map of hVDAC3 obtained by tryptic and chymotryptic digestion. The gray solid line indicates the sequence obtained with tryptic peptides; dotted lines the sequence obtained with chymotryptic peptides. The carboxyamidomethylated cysteine residues are highlighted in grey, and the partially oxidized cysteine residues are highlighted in orange. The partially oxidized methionine residues are highlighted in green. The acetyl group linked to the N-terminal cysteine is shown in red. Unique tryptic peptides originated by missed-cleavages were used to distinguish and cover sequences shared by isoforms: they are indicated in the boxes. Sequences that are shared by more isoforms are reported in bold. Numbering of the sequence considered the starting methionine residue, actually deleted during protein maturation

**Figure 6 ijms-21-01468-f006:**
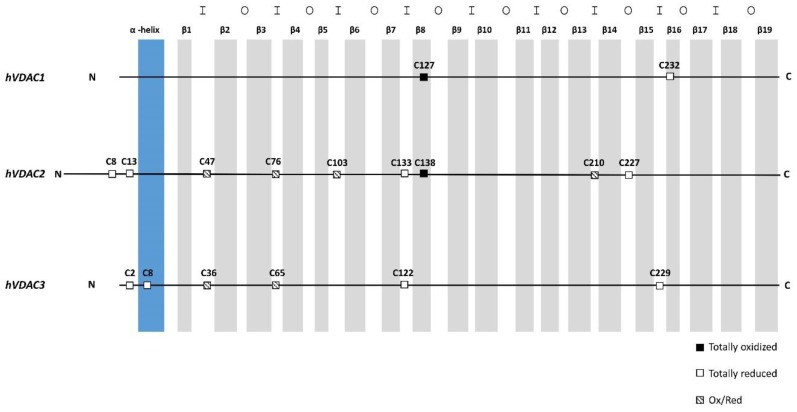
Scheme of the cysteine localization in the aligned sequences of human VDAC isoforms. N-terminal α-helix in blue, β-strands in gray. The internal loops, i.e., exposed to the inter membrane space, are indicated with I. The outer loops, i.e., exposed to the cytosol, are indicated with O.

**Table 1 ijms-21-01468-t001:** Comparison of the absolute intensities of molecular ions of selected sulfur containing tryptic peptides found in the analysis of hVDAC1 reduced with dithiothreitol (DTT), carboxyamidomethylated, and digested in-solution.

Peptide	Position in the Sequence	Measured Monoisotopic *m/z*	Absolute Intensity	Ratio Ox/Red
EHINLG**C**D**M**^129^DFDIAGPSIR	121–139	1083.9756 (+2)	1.6 × 10^5^	1.5
EHINLG**C**DMDFDIAGPSIR		1075.9784 (+2)	1.08 × 10^5^	
GALVLGYEGWLAGYQ**M**^155^NFETAK	140–161	812.0634 (+3)	6.4 × 10^5^	0.8
GALVLGYEGWLAGYQMNFETAK		1209.5944 (+2)	8.5 × 10^5^	

**M**: methionine sulfoxide; **C**: cysteine oxidized to sulfonic acid.

**Table 2 ijms-21-01468-t002:** Comparison of the absolute intensities of molecular ions of selected sulfur containing tryptic peptides found in the analysis of hVDAC2 reduced with DTT, carboxyamidomethylated, and digested in-solution.

Peptide	Position in the Sequence	Measured Monoisotopic *m/z*	Absolute Intensity	Ratio Ox/Red
P**M*****C***IPPSYADLGK	11–23	732.8476 (+2)	8.79 × 10^5^	9.8
PM***C***IPPSYADLGK		724.8497 (+2)	8.99 × 10^4^	
S**C**SGVEFSTSGSSNTDTGK	46–64	949.8815 (+2)	1.80 × 10^4^	0.01
S***C***SGVEFSTSGSSNTDTGK		954.4005 (+2)	1.59 × 10^6^	
YKW**C**EYGLTFTEK	73–85	858.3821 (+2)	1.53 × 10^4^	0.1
YKW***C***EYGLTFTEK		862.9037 (+2)	1.38 × 10^5^	
WNTDNTLGTEIAIEDQI**C**QGLK	86–107	837.3958 (+3)	1.09 × 10^5^	0.04
WNTDNTLGTEIAIEDQI*C*QGLK		1260.1086 (+2)	2.98 × 10^6^	
ECINLGCDVDFDFAGPAIHGSAVFGYEGW LAGYQ**M**TFDSAK^a^	132–172	1508.3225 (+3)	2.94 × 10^4^	1.4
ECINLGCDVDFDFAGPAIHGSAVFGYEGW LAGYQMTFDSAK^a^		1502.9921 (+3)	2.12 × 10^4^	
V**C**EDLDTSVNLAWTSGTN***C***TR	209–229	1195.5161 (+2)	4.40 × 10^4^	0.1
V***C***EDLDTSVNLAWTSGTN***C***TR		800.3594 (+3)	4.40 × 10^5^	

***C***: cysteine carboxyamidomethylated; **M**: methionine sulfoxide; **C**: cysteine oxidized to sulfonic acid. ^a^One of the two cysteines of these peptides is trioxidized and one carboxyamidomethylated, but it was not possible to determine which one was modified because the MS/MS spectrum was not obtained.

**Table 3 ijms-21-01468-t003:** Comparison of the absolute intensities of molecular ions of selected sulfur containing tryptic peptides found in the analysis of hVDAC3 reduced with DTT, carboxyamidomethylated, and digested in-solution.

Peptide	Position in the Sequence	Measured Monoisotopic *m/z*	Absolute Intensity	Ratio Ox/Red
GYGFG**M**VK	21–28	437.7098 (+2)	1.46 × 10^4^	0.06
GYGFGMVK		429.7118 (+2)	2.51 × 10^5^	
S**C**SGVEFSTSGHAYTDTGK	35–53	991.4069 (+2)	1.81 × 10^4^	0.2
S***C***SGVEFSTSGHAYTDTGK		995.9267 (+2)	8.80 × 10^4^	
YKV**C**NYGLTFTQK	62–74	806.8889 (+2)	1.01 × 10^4^	0.05
YKV***C***NYGLTFTQK		811.4081 (+2)	1.97 × 10^5^	
D***C***FSVGSNVDIDFSGPTIYGWAVLAFE GWLAGYQ**M**SFDTAK	121–161	1513.3688 (+3)	1.74 × 10^4^	1.1
D***C***FSVGSNVDIDFSGPTIYGWAVLAFE GWLAGYQMSFDTAK		1508.0270 (+3)	1.59 × 10^4^	
Y**M**LD***C***R	225–230	437.1832 (+2)	5.88 × 10^5^	3.2
YMLD***C***R		429.1861 (+2)	1.86 × 10^5^	

***C***: cysteine carboxyamidomethylated; **C**: cysteine oxidized to sulfonic acid; **M**: methionine sulfoxide.

**Table 4 ijms-21-01468-t004:** Comparison of the oxidation state of cysteines in rat and human VDAC isoforms. p. o.: partially oxidized; t.o.: totally oxidized; t.r.: totally reduced; rat: VDACs from rat liver; hu: VDACs from human HAP1 cells. Cysteine residues are aligned in the same rows according to their presumed homology.

Cysteine Position						Oxidative State					
VDAC1		VDAC2		VDAC3		VDAC1		VDAC2		VDAC3	
*rat*	*hu*	*rat*	*hu*	*rat*	*hu*	*rat*	*hu*	*rat*	*hu*	*rat*	*hu*
		4						t.r.			
		5						t.r.			
		9	8	2	2		-	t.r.	t.r.	t.r.	t.r.
		14	13	8	8		-	t.r.	t.r.	t.r.	t.r.
		48	47	36	36		-	p.o.	p.o.	p.o.	p.o.
		77	76	65	65		-	p.o.	p.o.	p.o.	p.o.
		104	103		-		-	p.o.	p.o.		-
		134	133	122	122		-	N.D.	t.r.	N.D.	t.r.
127	127	139	138		-	p.o.	t.o.	N.D.	t.o.		-
				165						p.o.	
		211	210		-		-	p.o.	p.o.		-
		228	227		-		-	p.o.	t.r.		-
				229	229		-		-	t.o.	t.r.
232	232				-	p.o.	t.r.		-		-

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
