# Peer review of "A High Resolution Mass Spectrometry Study Reveals the Potential of Disulfide Formation in Human Mitochondrial Voltage-Dependent Anion Selective Channel Isoforms (hVDACs)"

_ijms, 2020, doi:10.3390/ijms21041468_

Round 1

Reviewer 1 Report

Based on a detailed mass spectrometry analysis, this manuscript describes in detail the localization and oxidation states of cysteines as well as methionines in the three human porin isoforms. The implications of the different degree of such important posttranslational modification are relevant to get insights into the biological roles of these mitochondrial proteins. Based on their previous data of the rat isoforms, the authors also conclude that these proteins are highly conserved within mammals.

The data presented are convincing, supporting the conclusions reached. A few minor points follow:

There seems to be a problem with the References format that should be corrected. Thus, in the text, we see lowercase Roman numerals, most in superscript, whereas the list of References begins with Arabic numerals switching to Romans at Ref. xiv. Please rephrase the sentence beginning on pg. 2 – line 7, which is not clear, particularly, when saying “ii) there is a specific utilization of any specific cysteine, since same residues are always found in the reduced form”, which perhaps should be read as “ii) there is a no specific utilization of specific cysteines, since the equivalent same residues are always found in the reduced form”. For clarity, the residue equivalent to Glu72, which both hVDAC1 and hVDAC2 share, according to the authors, should be marked someway in Figures 1 and 4, respectively (see pg. 12 – line 38). In 4. Materials and Methods – 4.1. Chemicals, there are several abbreviations that likely belong to the list of standard abbreviations and thus may be used anywhere in the manuscript without further definition; the authors should verify this list in the journal guidelines and use directly the abbreviation if possible, e.g. for PBS (pg. 13 – line 19). Please specify that HAP1 cells are human when defined in 4. Materials and Methods (pg. 13 – line 24). Please rephrase the sentence beginning on pg. 15 – line 26: “The quest for the identification of disulfide bridge(s) is the next, highly challenging from the technical point of view, goal of this research.” It should say something like “From a technical point of view, the The quest for the identification of disulfide bridge(s) is the next, highly challenging from the technical point of view, goal of this our research.” Finally, given that Supplementary Figure S11 is visual, the authors should consider the possibility of including these diagrams in a main figure of the manuscript.

Author Response

Comments and Suggestions for Authors

Based on a detailed mass spectrometry analysis, this manuscript describes in detail the localization and oxidation states of cysteines as well as methionines in the three human porin isoforms. The implications of the different degree of such important posttranslational modification are relevant to get insights into the biological roles of these mitochondrial proteins. Based on their previous data of the rat isoforms, the authors also conclude that these proteins are highly conserved within mammals.

The data presented are convincing, supporting the conclusions reached. A few minor points follow:

There seems to be a problem with the References format that should be corrected. Thus, in the text, we see lowercase Roman numerals, most in superscript, whereas the list of References begins with Arabic numerals switching to Romans at Ref. xiv.

The reference format was corrected.

Please rephrase the sentence beginning on pg. 2 – line 7, which is not clear, particularly, when saying “ii) there is a specific utilization of any specific cysteine, since same residues are always found in the reduced form”, which perhaps should be read as “ii) there is a no specific utilization of specific cysteines, since the equivalent same residues are always found in the reduced form”.

The sentence was rephrased (see text pag.2, line 29).

For clarity, the residue equivalent to Glu72, which both hVDAC1 and hVDAC2 share, according to the authors, should be marked someway in Figures 1 and 4, respectively (see pg. 12 – line 38).

Glu72 (actually Glu73, considering that the sequence numbering  adopted  in the paper include the starting Met1, which is actually absent in the mature protein) in Figures 1 and 4 was marked by an asterisk and the corresponding figure captions modified accordingly. 

In 4. Materials and Methods – 4.1. Chemicals, there are several abbreviations that likely belong to the list of standard abbreviations and thus may be used anywhere in the manuscript without further definition; the authors should verify this list in the journal guidelines and use directly the abbreviation if possible, e.g. for PBS (pg. 13 – line 19).

The instruction for authors of  IJMS does not provide a list of standard abbreviation. For a better reading we added  a list of abbreviation used at the end of the paper.

Please specify that HAP1 cells are human when defined in 4.

In section 4.1 and 4.2 it was specified that HAP1 cells are human.

Materials and Methods (pg. 13 – line 24). Please rephrase the sentence beginning on pg. 15 – line 26: “The quest for the identification of disulfide bridge(s) is the next, highly challenging from the technical point of view, goal of this research.” It should say something like “From a technical point of view, the identification of disulfide bridge(s) is the next, highly challenging goal of this our research.”

The sentence was rephrased as suggested.

Finally, given that Supplementary Figure S11 is visual, the authors should consider the possibility of including these diagrams in a main figure of the manuscript.

We would prefer to not include Supplementary Figure S11 in the text in order to avoid excessive number of Figures.

Reviewer 2 Report

Pittala et al. focuses on the differential oxidation states among VDAC isoforms, based on primary sequence differences, and how these oxidation states may be relevant in cancer pathophysiology. The article, although highly specialized and focus in MS is interesting and worthy of publication in IJMS, provided certain aspects are improved/clarified:

Submission of the MS profiles to a public database is mandatory. The authors indicate that the submission is done (<PXDxxxxxx>), but I could not see the dataset anywhere in the manuscript nor in repositories. The level of oxidation is established as fold-change compared to basal conditions. Have the authors performed a “positive control” on oxidation, thus to treat the cells with oxidation agents, such as H2O2 or chloramine-T to determine the highest level of oxidation that the system can reach to set a high oxidation threshold? Have the authors performed the cell lysis with different detergents that Triton X-100, such as DDM or Digitonin, to determine that the results and/or VDAC solubilization rate are not affected by the experimental conditions? If the authors have done so, are there any differences among oxidation rates? This is probably an manuscript style issue, but the bibliography formatting using Roman numerals seems a bit odd.

Author Response

Comments and Suggestions for Authors

Pittala et al. focuses on the differential oxidation states among VDAC isoforms, based on primary sequence differences, and how these oxidation states may be relevant in cancer pathophysiology. The article, although highly specialized and focus in MS is interesting and worthy of publication in IJMS, provided certain aspects are improved/clarified:

Submission of the MS profiles to a public database is mandatory. The authors indicate that the submission is done (<PXDxxxxxx>), but I could not see the dataset anywhere in the manuscript nor in repositories.

The dataset was submitted to the ProteomeXchange Consortium and the dataset number has been added to the text.

The level of oxidation is established as fold-change compared to basal conditions. Have the authors performed a “positive control” on oxidation, thus to treat the cells with oxidation agents, such as H2O2 or chloramine-T to determine the highest level of oxidation that the system can reach to set a high oxidation threshold? Have the authors performed the cell lysis with different detergents that Triton X-100, such as DDM or Digitonin, to determine that the results and/or VDAC solubilization rate are not affected by the experimental conditions? If the authors have done so, are there any differences among oxidation rates?

The authors have performed extensive studies about the solubilization of mitochondrial VDAC with a wide range of detergents in a previous work (see De Pinto et al, Interaction of non-classical detergents with the mitochondrial porin  (1989) Eur. J. Biochem. 183, 179-187). This work clarified that the solubilization by Triton X-100 can be used as the Reference Procedure for obtaining optimal results because the detergent is harmless on the protein activity and is able to get the highest possible solubilization range. This last point is absolutely essential to proceed to the MS analysis where a significant amount of protein is required to perform a statistical meaningfull work. Thus the utilization of non ionic or natural detergents as DDM and Digitonin is not a useful comparison, since both have a lower solubilization power. In addition, they are both very expensive and their utilization is recommended for other purposes like mitochondria isolation or crystalization efforts.

Concerning the treatment with oxidation agents, it has been also performed in a previous work, but the oxidizing agents used were Cu-phenantroline or diamide, two reagents typically used in studies of mitochondrial protein redox situation (Reina et al. VDAC3 as a sensor of oxidative state of the intermembrane space of mitochondria: the putative role of cysteine residue modifications. Oncotarget. 2016 Jan 19;7(3):2249-68). Again, the treatment with the very strong oxidizing agents like H2O2 or cloramine T is able to destroy a physiological situation of the cell. The purpose of our work was exactly to monitor the VDAC cysteine modification without any external perturbation or limiting them as much as possible. 

This is probably an manuscript style issue, but the bibliography formatting using Roman numerals seems a bit odd.

The reference format was corrected.

Round 2

Reviewer 2 Report

Accept in present form. Although no new experiments have been performed, the author's explanations answer the referee main concerns.